# A Systematic Review of the Methods of Assessment of Gastro-Oesophageal Reflux in Anaesthetized Dogs

**DOI:** 10.3390/ani11030852

**Published:** 2021-03-18

**Authors:** Anna Carolina Fernandez Alasia, Olivier Levionnois, Mathieu Raillard

**Affiliations:** 1School of Veterinary Science, Faculty of Science, Evelyn Williams Building No B10, The University of Sydney, Sydney, NSW 2006, Australia; afer9691@uni.sydney.edu.au; 2Anaesthesiology Section, Department of Clinical Veterinary Sciences, Vetsuisse Faculty, University of Berne, 3012 Berne, Switzerland; olivier.levionnois@vetsuisse.unibe.ch

**Keywords:** anaesthesia, complications, dogs, gastro-oesophageal reflux, regurgitation, risk

## Abstract

**Simple Summary:**

Regurgitation and gastro-oesophageal reflux (GOR) are common complications in dogs under anaesthesia. We reviewed the definitions and methods of GOR assessment in anaesthetized dogs published in 22 scientific papers to assess if studies were comparable (i.e., looking at the same thing). The definition of GOR implied the presence of fluids not reaching the mouth or nose in the oesophagus in all studies. Most studies measured the acidity in the oesophagus to state if fluids were present or not. The probes were not always placed in the same location and definitions varied. This means that it is complicated to compare findings of the different studies.

**Abstract:**

We reviewed the definitions and methods of assessment of gastro-oesophageal reflux (GOR) in anaesthetized dogs. Three databases were used. Titles and abstracts were screened by two of the authors independently. A total of 22 studies was included in the analysis. The definition of GOR implied the presence of fluids not reaching the mouth or nose in the oesophagus in all studies. Most studies considered a change in pH using oesophageal pH meters as the sole method of assessment. Calibration of the pH probe was inconsistently reported. The position of the tip of the oesophageal probe was inconsistent and not always precisely described. The correct positioning in the intended location was verified in a limited number of studies. Some studies considered that GOR had happened for changes in pH below 4.0 or above 7.5 while others considered that GOR had happened when the pH dropped below 4.0 only. Some studies stated that the pH change had to be sustained for a minimum period of time (20 or 30 s) whereas others did not mention any duration. The variability of definitions and methods of assessment of GOR in anaesthetized dogs precludes meaningful comparison of the findings. Re-evaluation and uniformization of the methods appear necessary.

## 1. Introduction

Regurgitation and gastro-oesophageal reflux (GOR) are common complications in dogs undergoing general anaesthesia and can lead to significant morbidity and mortality. The literature highlights several risk factors (i.e., age, body weight, and type of surgery) and reports that the incidence of regurgitation and GOR may be influenced by a number of interventions (i.e., pre-operative fasting, positioning, and drugs). Reported incidences seem to vary reasonably for regurgitation [from 0.96% [1] to 5.5% [2],] but enormously for GOR [from 5% [3] to 87.5% [4]]. Such a huge variability is rather surprising. Before contrasting published findings and interventions aimed to reduce the development of regurgitation and GOR, it seems legitimate to question the methods used in the scientific literature.

The aim of this study was to review the definitions and methods of assessment of GOR in anaesthetized dogs in clinical veterinary practice.

## 2. Materials and Methods

The PRISMA (Preferred Reporting Items for Systematic Reviews and Meta-Analyses) checklist was used.

The review protocol was not registered. The search was electronic. The search strategy was as follows: ((dogs OR canines OR dog OR canine) AND (anaesthetized OR anesthetized OR anaesthetize OR anesthetize OR anaesthetise OR anaesthesia OR anesthesia) AND (gastro-oesophageal reflux OR GER OR GOR OR gastroesophageal reflux or gastro-oesophageal reflux)). Three databases were used: Pubmed, Embase, and Scopus. The search was last performed on the 6 December 2020. References of relevant publications were also consulted. Titles and abstracts were screened by two of the authors independently. There was no *a priori* year restriction. Case reports, case series, conference papers. and non-English publications were excluded. The focus of this review being anaesthesia in clinical veterinary practice, studies including dogs as an experimental model and studies not focusing on GOR during the intra-anaesthetic period were excluded.

Information extracted included: (1) definitions and endpoints used in the article for regurgitation; (2) definitions and endpoints used in the article for GOR; (3) methods of assessment of regurgitation and GOR; (4) pH probe calibration if appropriate (method and timing); (5) equipment positioning, time of insertion, and removal; (6) verification of the appropriate location of the probe and timing of the check; (7) frequency of measurements; and (8) particular precautions (to limit probe dislodgement or effort to limit the iatrogenic GOR and regurgitation).

Descriptive statistics were performed where appropriate.

## 3. Results

The consort diagram for the search strategy is presented in Figure 1. A total of 164 studies were assessed for eligibility: 76 in Pubmed (56 were excluded: 54 not related to the question, one non-primary research, one abstract), 69 in Embase (48 were excluded: 43 not related to the question, two non-primary research, three abstracts); 19 in SCOPUS (9 were excluded as not related to question). Duplicates were removed. A total of 22 submissions was reviewed. 

The 22 manuscripts assessed in this review are reported in Table 1. A total of 10/22 studies did not talk about regurgitation. The definition of regurgitation generally implied the passive and visible discharge of fluid from the mouth or nose. However, one study considered a change in pH in the pharynx as an episode of regurgitation. In that study, pharyngeal pH was measured when GOR episodes were identified, to evaluate the spread of the reflux.

The definition of GOR implied the presence of fluids not reaching the mouth or nose in the oesophagus in all studies. The portion of the oesophagus considered was infrequently reported in the definition: “distal” or “caudal” or “lower” oesophagus in 8/22 papers. Most studies (20/22) identified the development of GOR through a change in oesophageal pH. This was the sole method of assessment in 18/22 studies, while two studies used oesophagoscopy on top of the pH meter to visualise if reflux was present (i.e., non-acid reflux not causing pH changes). One study used a combined pH/impedance probe and considered a 50% decrement in Ohms seen in two consecutive impedance channels in the distal oesophagus for >2 s compared with the pre-episodic oesophageal baseline recording. One study retrospectively assessed the presence of gas, fluid, or alimentary content on CT (computed tomography) images.

In the 20 studies considering pH changes, 14 considered that GOR had happened for changes in pH “below 4.0 or above 7.5” while six considered that GOR had happened when the pH dropped below 4.0 only. The study using impedance changes categorized the pH of the reflux but this was not the criteria used to state if GOR had happened. In addition, for GOR to be confirmed, six studies stated that the pH change had to be sustained for a minimum period of time (20 or 30 s) whereas the others did not mention any duration in pH change.

Irrespective of definitions and main outcome measures, 21/22 of the studies used pH meters. Calibration was reported in only 14 studies. Two-point calibration (generally with buffers of pH 1 or 4 and pH 7) was used.

The position of the tip of the oesophageal probe was inconsistent and not always described with precision. The distance between the lower incisors and the cranial margin of the 10th rib was externally measured with the dogs in lateral recumbency in 14 studies. However, the probe was advanced that distance in only 10/14 studies, whereas it was advanced that distance minus 5 cm in 4/14. The distance between the upper canine and the distal border of the ninth rib (in a straight line with the dogs’ head and neck in a “normal position”) was measured in two studies where the probe was advanced that distance. It was estimated to position the tip of the probe about 7 cm cranially to the lower oesophageal sphincter in a “preliminary trial” where radiography was used. The probe was advanced until the ninth rib in one study. It was retrieved from the stomach, 6 cm cranially to the lower oesophageal sphincter (visualised through video-oesophagoscopy) in one study. The pH was measured at three different levels in the oesophagus (thoracic inlet, fifth and ninth rib) in one study. The location of the probe placement was not reported in 2/21 studies. The correct positioning of the probe in the intended location was verified through endoscopy, chest radiographs, or fluoroscopy in only 5/21 studies. The permanence of the probe in the appropriate location at the beginning and at the end or throughout the study was never checked.

The full results are presented in Appendix A.

## 4. Discussion

A variety of definitions and methods of assessment of GOR in anaesthetized dogs is present in the literature. Although oesophageal pH measurement is the method most commonly used, calibration, position of the probe, and cut-off values of pH to differentiate between GOR or not are inconsistent.

The method of analysis of pH is relative. This means that pH meters need to be appropriately calibrated [24]. The pH calibration curve is a combination of two curves, the pH and the pOH curves and is not linear. A slight deviation in the range of pH 6–8 is expected [24]. Generally, a pH meter should be calibrated every 2 to 3 h using at least two buffer solutions with known pH values close to the expected pH to be measured [24]. When considering a wide range of pH, two-point calibration is not sufficient [24]. Also, given the effects of temperature on pH measurements [24], the question of calibrating the instruments at body temperature exists. In the present review, several studies used oesophageal pH meters without reporting any calibration. Studies reporting calibration used a two-point calibration, mostly using buffers of 1 or 4 and 7 while eight studies were defining GOR as a pH change below 4 or above 7.5. The accuracy of relevant pH values presented is questionable. Each pH unit change represents a 10-fold change in the hydrogen or hydroxyl ions concentration. The calibration might be less important if definitions considered sudden changes in pH as a proxy biomarker of GOR instead of numerical cut-off values.

Anatomic landmarks were commonly used to estimate the position of the lower oesophageal sphincter and the length of the probe to advance in the oesophagus. Most studies used the description proposed by Waterman and Hashim [25]. However, the length of the oesophageal probes advanced was variable. Also, the position of the tip of the probe was rarely checked. Depending on the volume of the refluxate, actual GOR could be missed, for example, in the case of pH measurements in the proximal oesophagus if material is present only caudally or if the tip of the probe is not in the liquid phase in the dependent oesophagus. Furthermore, there was no consensus on the definition of GOR and pH cut-off values of the refluxate. Although acid reflux seems to be observed more frequently than alkaline reflux in the articles examined, actual incidence of GOR might be underestimated in some studies. These points make comparisons between studies challenging from the perspective of actual incidence or efficacy of measures taken.

Consequences of GOR can include oesophagitis, oesophageal stricture, and aspiration pneumonia [26,27]. Evidence in humans suggests that mixed reflux (acid mixed with bile acids) is more harmful to the oesophageal mucosa than acid reflux alone [28] and that bile reflux in the oesophagus can occur over a wide range of pH (2–8) [29]. Although the actual importance of this information in dogs undergoing a single anaesthesia and GOR episode is not known, the clinical relevance of changes in oesophageal pH as a sole marker of GOR is questionable. The position of the tip of the probe as well as the cut-off values of pH used to assess the presence vs absence of GOR might require re-evaluation. Although pH-metry seems commonly used, likely because of its wide availability and low cost, this technique alone might be insufficient. Combined pH/impedance probes with multiple impedance channels to evaluate the spread of the refluxate in the oesophagus, similar to the product used by Zacuto et al. (2012) [16] or Tarvin et al. (2016) [30], calibrated and in a well identified (checked) location, associated with biological analysis of the refluxate might offer a more relevant picture of GOR in anaesthetized dogs.

## 5. Conclusions

The variability of the GOR incidence found in the literature is likely due to a variety of factors (i.e., anaesthetic depth, transport, and position). However, the multiple definitions and methods of assessment of GOR in anaesthetized dogs present in the literature preclude meaningful comparison of the findings. Re-evaluation and uniformization of the methods seem necessary. Some aspects might warrant further investigation: (1) the relevance of the volume of the material regurgitated and how long the reflux remains in the oesophagus for; (2) the impact of anaesthesia on oesophageal motility; and (3) the importance of anaesthetic depth and the presence of monitoring equipment in the oesophageal lumen on the incidence of GOR.

## Figures and Tables

**Figure 1 animals-11-00852-f001:**
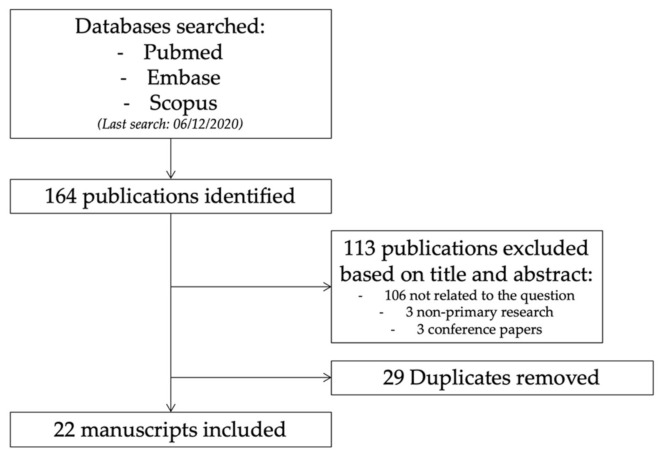
Consort diagram for search strategy.

**Table 1 animals-11-00852-t001:** Reference number, authors, year of publication, and journal of the 22 manuscripts included in the review of the methods of assessment of gastro-oesophageal reflux (GOR) in anaesthetized dogs.

Ref.	Authors	Year	Journal
[5]	Roush JK, Keene BW, Eicker SW et al.	1990	*Vet. Surg.*
[6]	Galatos AD, Raptopoulos D	1995	*Vet. Rec.*
[7]	Galatos AD, Raptopoulos D	1995	*Vet. Rec.*
[2]	Wilson DV, Evans AT, Miller R	2005	*Am. J. Vet. Res.*
[8]	Wilson DV, Boruta DT, Evans AT	2006	*Am. J. Vet. Res.*
[9]	Wilson DV, Evans AT, Mauer WA	2006	*Am. J. Vet. Res.*
[10]	Wilson DV, Tom Evans A, Mauer WA	2007	*Vet. Anaesth. Analg.*
[11]	Wilson DV, Evans AT	2007	*Vet. Anaesth. Analg.*
[12]	Anagnostou TL, Savvas I, Kazakos GM et al.	2009	*Vet. Anaesth. Analg.*
[13]	Panti A, Bennett RC, Corletto F et al.	2009	*J. Small Anim. Pract.*
[14]	Favarato ES, de Souza MV, dos Santos Costa PR et al.	2011	*Vet. Res. Commun.*
[15]	Favarato ES, Souza MV, Costa PR et al.	2012	*Res. Vet. Sci.*
[16]	Zacuto AC, Marks SL, Osborn J et al.	2012	*J. Vet. Intern. Med.*
[17]	Johnson RA	2014	*Vet. Anaesth. Analg.*
[18]	Anagnostou TL, Savvas I, Kazakos GM et al.	2015	*Vet. Anaesth. Analg.*
[3]	Savvas I, Raptopoulos D, Rallis T	2016	*J. Am. Anim. Hosp. Assoc.*
[19]	Anagnostou TL, Kazakos GM, Savvas I et al.	2017	*Vet. Anaesth. Analg.*
[20]	Shaver SL, Barbur LA, Jimenez DA et al.	2017	*J. Am. Anim. Hosp. Assoc.*
[21]	Torrente C, Vigueras I, Manzanilla EG et al.	2017	*J. Vet. Emerg. Crit. Care*
[22]	Viskjer S, Sjostrom L	2017	*Am. J. Vet. Res.*
[4]	Lambertini C, Pietra M, Galiazzo G et al.	2020	*Vet. Sci.*
[23]	Benzimra C, Cerasoli I, Rault D et al.	2020	*J. Vet. Sci.*

## Data Availability

Data is contained within the article.

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
