# Peer review of "A Systematic Review of the Methods of Assessment of Gastro-Oesophageal Reflux in Anaesthetized Dogs"

_animals, 2021, doi:10.3390/ani11030852_

Round 1
Reviewer 1 Report
Dear Authors,
This is an excellent review.
I have very few comments.
Thanks for this excellent work.
L61 : remove « irrelevant »
L71 : just mention you did descriptive statistics. It’s sufficient.
L 85 and L90 : I would prefer to see the definition of reflux and GOR in the introduction.
L 134-136 : references are missing
L141-144 : this is a result, so this should be in the result part.
L147 : I don’t understand the sentence « however…in the definitions »
L153 : you should clarify the GOR is most likely to be detected in the caudal esophagus if this is what you mean. Also, the lumen of the esophagus as a certain volume. How can we be certain the probe is located in the dependent position of the esophagus, where the GOR can be observed ?
L171 : could you mention which method could be used to check the localization of the probe
L179 : « the duration of the permanence « : this is unclear
Author Response
Dear Reviewer,
Thank you for your kind and constructive review. We have adjusted most of your concerns, please find our comments below. We hope this will match your expectations.
Best wishes,
The authors
L61 : remove « irrelevant »
Irrelevant has been removed.
L71 : just mention you did descriptive statistics. It’s sufficient.
Amended as requested.
L 85 and L90 : I would prefer to see the definition of reflux and GOR in the introduction.
We would prefer to let this section in the results as the evaluation of the definitions was the primary focus of this review.
L 134-136 : references are missing
Reference added as requested.
L141-144 : this is a result, so this should be in the result part.
The information in the discussion re-uses what was already reported in the results. Wording has been altered to fit better the discussion part. We hope the reviewer will be happy with this amendment.
L147 : I don’t understand the sentence « however…in the definitions »
The sentence has been reworded.
L153 : you should clarify the GOR is most likely to be detected in the caudal esophagus if this is what you mean. Also, the lumen of the esophagus as a certain volume. How can we be certain the probe is located in the dependent position of the esophagus, where the GOR can be observed ?
Thank you for this comment; the sentence has been modified and clarified.
L171 : could you mention which method could be used to check the localization of the probe
We believe this problem should be adjusted by the authors of future research depending on what innovative technique of assessment of GOR they go for.
L179 : « the duration of the permanence « : this is unclear
The sentence has been clarified.Reviewer 2 Report
This is a well performed study and a well written manuscript which is of great importance to many a kind of people! Although it is well known, probably not only to researchers studying GOR but also to readers of their papers, that there are many inconsistencies between various studies, it is of great importance that a manuscript points out at least some of them! It is my opinion that this will enable not only researchers to design and perform better studies in the near future and thus to collect important data that are meaningful and readily comparable to the data of other studies, but also, hopefully, will help editors and reviewers of such manuscripts... This is not in the least less important because, in my opinion, researchers and authors may be responsible for improperly designed studies and written manuscripts, but the blame for "scientific pollution" should be put on editors and reviewers...
Despite the fact that the authors chose to review only the definitions and methods of assessment of gastro-oesophageal reflux in anaesthetized dogs, ignoring other factors [to mention just a few: a) exclusion criteria, such as vomiting after premedication, b) proper definition of terms, such as "sufficiently deep plane of anaesthesia to tolerate insertion", c) reference to conditions, such as "probe taped to an oesophageal stethoscope advanced (...) into the oesophagus " or "visualisation of content through video-oesophagoscopy", whose impact may render almost worthless a rather well designed study, or d) not reference to conditions, such as "No transportation of the animals to another operation room" ] that may be even more important for extracting or, actually, not extracting meaningful and comparable data, this remains a manuscript that certainly warrants publication. After all, this fine study may be followed in the near future by a second one taking into consideration the aforementioned factors...
Finally, I would like to mention the fact that, as it seems, none of the authors has published a study on GOR during anesthesia so far. Although this may be considered a disadvantage at first sight, it is my opinion that, on the contrary, it is rather an advantage simply because it permits them to take a fresh look at the matter, which wouldn't be easy for other researchers. I feel obliged to thank them for their impact on the study of GOR during anaesthesia.
Author Response
Dear Reviewer,
Thank you for your review. We do appreciate your kind comments and take good note of your suggestion for future work.
Best wishes,
The authors
Reviewer 3 Report
Well done!